# Vascular Inflammatory Diseases and Endothelial Phenotypes

**DOI:** 10.3390/cells12121640

**Published:** 2023-06-15

**Authors:** Jenita Immanuel, Sanguk Yun

**Affiliations:** Department of Biotechnology, Inje University, Gimhae-si 50834, Republic of Korea; jenitaimmanuel@gmail.com

**Keywords:** endothelial cell, inflammation, EC phenotypes, vascular diseases

## Abstract

The physiological functions of endothelial cells control vascular tone, permeability, inflammation, and angiogenesis, which significantly help to maintain a healthy vascular system. Several cardiovascular diseases are characterized by endothelial cell activation or dysfunction triggered by external stimuli such as disturbed flow, hypoxia, growth factors, and cytokines in response to high levels of low-density lipoprotein and cholesterol, hypertension, diabetes, aging, drugs, and smoking. Increasing evidence suggests that uncontrolled proinflammatory signaling and further alteration in endothelial cell phenotypes such as barrier disruption, increased permeability, endothelial to mesenchymal transition (EndMT), and metabolic reprogramming further induce vascular diseases, and multiple studies are focusing on finding the pathways and mechanisms involved in it. This review highlights the main proinflammatory stimuli and their effects on endothelial cell function. In order to provide a rational direction for future research, we also compiled the most recent data regarding the impact of endothelial cell dysfunction on vascular diseases and potential targets that impede the pathogenic process.

## 1. Introduction

Endothelial cells line the luminal surface and act as a barrier separating blood and surrounding tissue. Its dynamic and heterogeneous structure influences various important processes, such as vascular permeability, homeostasis, angiogenesis, metabolism, inflammatory cell trafficking, vasomotor tone, and immunity [1,2,3]. Remarkably, its well-defined barrier structure prevents extravasation of liquids, ions, chemicals, and leukocytes, and signaling pathways are functionally aligned according to the demands under certain conditions. Intercellular junctions regulate endothelial permeability with the help of junctional protein complexes called adherens junctions (AJs), gap junctions (GJs), tight junctions (TJs), and other adhesion receptors, such as platelet-endothelial cell adhesion molecule-1(PECAM-1) [4,5,6]. In regulating vascular tone, the endothelium plays a crucial role by releasing a variety of relaxing factors, including nitric oxide (NO), endothelium-dependent hyperpolarization factors, and vasodilator prostaglandins. NO is a critical component of a healthy vascular endothelium and helps keep the vascular wall in a quiescent state by preventing thrombosis, cellular proliferation, and inflammation. This quiescent, NO-dominated endothelium phenotype is most likely maintained by laminar shear stress [7,8].

Vascular disease is a result of endothelial dysfunction, which is often referred to as endothelial activation in pathological situations. A change from a quiescent phenotype to one that engages the host defense response is represented by endothelial activation. In fact, the majority of cardiovascular risk factors trigger endothelium-based molecular machinery, causing the expression of chemokines, cytokines, and adhesion molecules that are intended to interact with leukocytes and platelets and target inflammation in specific sites to eliminate pathogens. The basic change occurring in this process is a shift in the signaling from NO-mediated silencing of cellular processes to redox-mediated activation through reactive oxygen species (ROS) [9]. It is noteworthy that NO, which generally aids in retaining the endothelium in a quiescent state, can be converted to ROS under certain conditions as part of endothelial activation, which is called eNOS uncoupling. Endothelial functions are crucial for ensuring the appropriate maintenance of vascular homeostasis. Depending on the type, severity, duration, and combination of the proinflammatory stimuli, endothelial activation and redox signaling may promote host defense or trigger vascular inflammatory diseases. Increasing evidence suggests that endothelial dysfunction is a hallmark of a variety of cardiovascular conditions associated with pathological states including vasoconstriction, leukocyte adhesion, thrombosis, and inflammatory state.

The overall aim of this review is to summarize the latest research on the contribution of ECs to vascular inflammatory diseases. To this end, endothelial phenotypes and stimuli will be discussed with regard to EC-specific shear stress signaling. Finally, several important vascular diseases will be discussed in relation to the EC inflammatory phenotypes. Since a couple of thorough reviews on endothelial cell-focused vascular diseases not many up-to-date reviews on endothelial cells in vascular diseases are available. Especially, with remarkable progress in NGS sequencing technology, a lot of new data on endothelial phenotypes have been accumulated, so it would be timely to compile key progresses on endothelial phenotype analysis. This review first focuses on endothelial phenotypes underlying cell signaling and then shifts to address how endothelial phenotypes contribute to individual diseases.

## 2. Endothelial Proinflammatory Phenotype

Endothelial cells in healthy resting vasculatures contribute to vascular homeostasis by keeping vascular inflammation, thrombosis, and permeability low. Under various EC-activating stimuli, endothelial cells undergo phenotypic changes to meet the needs for immune cell recruitment or new vessel formation. During infection, endothelial cells exhibit proinflammatory phenotypes to fight pathogens. However, uncontrolled inflammation can damage host tissue and lead to various vascular inflammatory diseases. The phenotypic changes occurring in ECs during vascular inflammation are discussed below.

### 2.1. Cell Surface Adhesion Molecule Expression

Endothelial inflammatory signaling triggered by inflammatory stimuli leads to the expression and cell surface localization of cell adhesion molecules (CAMs). The main function of CAMs is to facilitate leukocyte infiltration from the blood stream into the inflamed tissues. Vascular cell adhesion molecules-1 (VCAM-1) and intercellular adhesion molecules-1 (ICAM-1) are well-known transcriptional targets of nuclear factor kappa-light-chain-enhancer of activated B cells (NF-κB) and Yes-associated proteins (Yap) [10]. Their surface expressions are upregulated in ECs during inflammation and involved in leukocyte adhesion and rolling [11]. JAM molecules (JAM-A, JAM-B, and JAM-C) are mediators of leukocyte–EC and platelet–EC interaction via heterophilic trans-interactions [12]. Under resting conditions, homophilic trans-interactions of JAM-A contribute to TJ formation. However, under inflammatory conditions, junctional JAM-A mobilizes to the apical side for leukocyte binding. Selectins are a family of three C-type lectins expressed by bone-marrow-derived cells and endothelial cells [13]. Selectins have a carbohydrate recognition domain that binds specific glycans on leukocytes [14]. ECs express E-selectin and P-selectin. Chemokines are low-molecular-weight peptides that often act as chemotactic factors in inflammation, and they comprise four groups (CXC, CC, C, and CX3C) depending on the number and spacing of conserved cysteines. ECs secrete numerous chemokines and also express chemokine receptors on their surfaces [15]. Ox-LDL or disturbed flow induce monocyte chemoattractant protein-1 (MCP-1) expression on the vascular wall, which mediates monocyte recruitment and atherosclerosis through the interaction between CCL2 on MCP-1 and CCR2 on monocytes [16].

### 2.2. Increased Endothelial Permeability

Endothelial permeability can be categorized into paracellular permeability and transcellular permeability. Paracellular permeability increases when tight junction (TJ) or adherens junction (AJ) mediated by homophilic adhesion molecules such as VE-cadherin, Occludin, or Claudin are disrupted. Transcellular permeability refers to the passage of molecules (transcytosis) or immune cells (transcellular migration) across plasma membranes via endocytic pathways. Under neuroinflammation, Th1 cells use a transcellular migration pathway, whereas Th17 cells use paracellular pores formed by TJ remodeling for CNS entry [17]. Another finding has shown that CNS endothelial cells have a specific mechanism inhibiting caveolae-mediated transcytosis, which promotes BBB integrity [18]. EC barrier function is compromised in various vascular inflammatory conditions. In infections, an increase in permeability is required for the recruitment of circulating leukocytes to the tissue area via a process called diapedesis [19]. Under pathological conditions, uncontrolled inflammation and vascular leakage often become detrimental due to fluid leakage into tissues, inducing edema and organ dysfunction. Inflammatory insults lead to VE-cadherin-mediated cell–cell junction disruption by src-dependent tyrosine phosphorylation [19] and the internalization of VE-cadherin. Disassembled junctions promote transendothelial migration (TEM) of monocytes to the subendothelial area [20]. A molecular weight of greater than 50,000 prevents extravasation under resting conditions, but not under inflammation or tumors [21,22] In early atherosclerosis, LDLs accumulate in the subendothelial region via transcytosis [23,24]. Fibrinogen extravasation in an inflamed Alzheimer’s disease brain contributes to neuronal damage in AD patients [25].

### 2.3. EndMT

A recent analysis revealed that atherosclerotic plaque has EC-derived mesenchymal cells, which increase in number as atherosclerosis progresses [26]. In vivo lineage tracing experiments showed that approximately 30% of aortic ECs express mesenchymal markers NOTC3 or FAP in hyperlipidemia mouse models via EndMT process [27,28]. Endothelial-to-mesenchymal transition (EndMT) refers to the transition of endothelial cells into less-differentiated mesenchymal cell types [29]. EndMT involves the loss of cell–cell junction, fibroblast-like cell morphology, and fibrosis [26]. Inflammatory ligands IL1b, TGF-β, and TNF-𝛼 and disturbed flow are known EndMT-inducing stimuli. Endothelial cells that endure EndMT cease to express EC-specific proteins such as vascular epidermal growth factor receptor (VEGFR), CD31/(PECAM-1), and vascular endothelial cadherin (VE-cadherin) and begin to express and produce mesenchymal-cell-specific proteins such as α-smooth muscle actin (α-SMA), vimentin, fibronectin, N-cadherin, fibroblast-specific protein-1 (FSP-1), fibroblast activating protein (FAP), and fibrillar collagens. Numerous studies have demonstrated that the TGF-β family of growth factors are the primary inducers of EndMT. However, EndMT is a highly complex process that involves a wide range of TGF-β and non-TGF-β signaling pathways and is regulated by a number of molecular processes depending on the health or pathological status of the cells as well as their unique cellular environment. TGF-β signaling mainly induces EndMT through SMAD 2/3 phosphorylation. Several other non-TGF-β signaling mechanisms, such as MAPK, PI3K, and PKC-δ, also induce EndMT. Previous studies reported that oscillatory shear stress promotes the activation of BMP, FGF, NOTCH, WNT, and ET-1, increasing mesenchymal markers through transcription factors SNAI1 and TWIST [30,31].

### 2.4. Senescence

Cellular senescence is a state of permanent cell cycle arrest due to various stresses [32]. Inflammation, oxidative stress, and disturbed flow have been known to induce premature endothelial senescence [32,33]. Senescence leads to endothelial dysfunction and arterial stiffness, thus contributing to cardiovascular diseases such as atherosclerosis.

## 3. EC Proinflammatory Stimuli

Endothelial proinflammatory phenotypes are induced by various stimuli. Some inflammatory ligands elicit fast and transient EC responses (type I endothelial activation), and others mediate more sustained EC responses involving new gene inductions (type II endothelial activation). The vast majority of inflammatory ligands acting on ECs also act on leukocytes by inducing the same signaling pathways. Shear stress seems to be an EC-specific inflammatory stimulus. Here, we present an overview of ligand-type EC stimuli and non-conventional proinflammatory stimuli.

### 3.1. Type I Endothelial Activation Ligands

Endothelial activation was originally used to refer to the quantitative changes in the level of protein expression that endow endothelial cells with new endothelial functions [34]. Endothelial activation was later grouped into two stages depending on the extent of the involvement of protein synthesis. Type I endothelial activation does not require de novo protein synthesis or gene upregulation and occurs rapidly. Binding a ligand such as histamine or thrombin to the GPCR induces heterotrimeric G-protein and subsequently PLC-beta to release intracellular calcium. In another arm, Rho is activated by Gho Gef, which is activated by Gbr subunits. Calcium-activated MLCK and Rho kinase together regulate actin cytoskeletal contraction, open EC junctions, and induce exocytosis of WPB (Weibel–Palade bodies), bringing P-selectin to the luminal surface to enhance leukocyte recruitment [35].

### 3.2. Type II Endothelial Activation Ligands

Cytokines are a group of soluble glycoproteins and peptides produced by immune cells and vascular cells [36]. Well-known proinflammatory cytokines stimulating ECs include IL-1β, TNF-α, TGF-β, and IL-6. IL-1β is produced by monocytes, macrophages, and ECs [37]. EC-induced IL-1 β production is mediated by the NLRP3 inflammasome and pyroptosis pathway [38,39]. IL-1 β is involved in host defenses and the pathogenesis of vascular diseases via inducing prostaglandin production, iNOS induction, the induction of many other cytokines and leukocyte adhesion molecules, and the activation of cells involved in innate immunity [40,41]. Kondreddy et al. recently revealed the mechanism underlying IL-1β-induced inflammation and thrombosis, which involves Gab2 regulation of the CBM signalosome assembly and rho and NF-κB activation [42] in deep-vein thrombosis (DVT). Monocytes and macrophages are the main TNF-α-producing cells. Lipopolysaccharide (LPS) stimulation of macrophages induces high TNF-α production [43]. Acute TNF-α leads to shock, tissue injury, and vascular damage, whereas chronic low levels of TNF-α induce anorexia, protein catabolism, lipid deletion, insulin resistance, and endothelial activation. TNF receptor 1 (TNFR1) is ubiquitously expressed, while TNF receptor 2 (TNFR2) is limited to hematopoietic and endothelial cells. Recently, EC-derived TNF-α has been shown to be required for tumor metastasis [44]. TNFR1 ectodomain shedding by ADAM17 induces endothelial necroptosis, leading to tumor cell extravasation. All cells in the arterial wall can produce TGF-β [45] in vascular diseases, exerting a diverse range of actions on each cell involved. EndMT is induced through EC exposure to TGF-β, which binds TGFBR2 and ALK2 or ALK5 to induce SMAD2/3/4 complexes via interactions with Snail1, Snail2, Zeb1, Zeb2, KLF4, TCF3, and Twist to induce mesenchymal transcription factors [46]. Endothelial responses to TGF-β include extracellular matrix (ECM) production and EndMT, a process by which ECs lose their identity and instead adopt a mesenchymal/myofibroblastic character. TGF-β signaling is balanced by BMP signaling. Hiepen et al. demonstrated that the BMP type 2 receptor (BMPR2) serves as a gatekeeper of this balance and BMPR2 deficiency leads to the formation of mixed heteromeric receptor complexes comprising BMPR1/TGFβR1/TGFβR2, enhancing TGF-dependent EndMT phenotypes [47]. IL-6 is released in the innate immune response by leukocytes as well as stromal cells upon pattern recognition receptor activation [48]. IL-6 induces acute-phase proteins, C-reactive protein (CRP), several complement system proteins, and the coagulation cascade. IL-6 released from muscles during exercise triggers anti-inflammatory responses [49]. An elevated level of IL-6 is a hallmark of cytokine storms [50]. IL6 elicits endothelial injury via VE-cadherin disassembly and increased endothelial C5a receptor expression, which also promotes vascular leakage.

### 3.3. High Glucose

Vascular inflammation is highly correlated with diabetic conditions [51]. Circulating inflammatory cytokines IL-1β, TNF-α, and IL-6 are increased in type 2 diabetes [52]. Hyperglycemia induces ROS formation in ECs and NF-κB activation. High circulating glucose promotes the TCA cycle and production of electron donors derived from the electron transport chain [53]. Hyperglycemia elevates protein O-GlcNAcylation via the hexosamine biosynthetic pathway and modifies the NF-κB pathway [54]. Westuck et al. recently reported that the proatherogenic effect of hyperglycemia is due to endothelial activation and the increased number of cell adhesion molecules such as P-selectin, E-selectin, and VCAM-1 [55].

### 3.4. Ox-LDLs

A total of 40–60% of serum LDLs are eliminated by hepatic LDL receptors in the liver. Macrophage-dependent uptake of oxidized LDLs is mediated by scavenger receptors, leading to foam cell formation in atherosclerotic lesions [56]. LDL oxidation occurs on the arterial wall via metal ions, lipoxygenase, xanthine oxidase, and myeloperoxidase, involving several pathways [57]. LOX-1 is an ox-LDL receptor on endothelial cells. Upon ox-LDL binding, LOX-1 triggers a plethora of signaling pathways in inflammation, apoptosis, oxidative stress, vasomotion, and angiogenesis. LOX-1 is implicated in multiple vascular diseases, including atherosclerosis, myocardial infarction, stroke, restenosis, thrombosis, and so on [58].

### 3.5. PAMP and DAMP

The host defense system can recognize the molecular components of invading pathogens, which are called pathogen-associated molecular patterns (PAMPs) [59]. Lipopolysaccharide (LPS), a component of the outer membrane component of Gram-negative bacteria, is a well-known PAMP. PAMPs lead to the release of proinflammatory cytokines and endogenous molecules called DAMPs (danger-associated molecular patterns) during cell injury processes. HMGB-1 is a DAMP released by injured endothelial cells in response to inflammation and infection that promotes inflammatory cytokine production, progression of atherosclerosis, and plaque vulnerability [60]. Various LPS-induced proinflammatory signaling pathways are mediated by Toll-like receptor 4 (TLR4) on endothelial cell surfaces. However, it was recently reported that LPS can enter the endothelial cytoplasm via bacterial microvesicles or bacterial breaching of the endothelial plasma membrane and that this intracellular LPS can induce endothelial pyroptosis via caspase-4/5/11-mediated inflammasome pathway activation [61]. The formation of gasdermin D (GSDMD)-mediated membrane pores allows an extracellular release of IL-1β or DAMPs and eventual cell lysis, further aggravating inflammation [62]. 

### 3.6. ECM (Extracellular Matrix)

Subendothelial ECM composition is altered during vascular development, wound healing, and inflammation. Upregulation of fibronectin is associated with various vascular diseases, including atherosclerosis, aneurysm, and psoriasis [63,64,65]. Integrin activation occurs under mechanical stimuli such as shear stress, cyclic stretch control, and EC signal transduction and phenotypes. Several soluble ligands, including ox-LDL and IL1β, which induce inflammatory responses, are also influenced by ECM–integrin couplings [66].

### 3.7. Disturbed Flow

EC phenotypes are profoundly affected by blood profile. A disturbed flow profile in the curvature or bifurcation of arteries can induce ROS formation and increase vascular permeability and immune cell adhesion molecule upregulation. EC mechanotransduction in blood flow sensing is implicated in various vascular diseases. For example, atherosclerotic plaques begin forming in the atheroprone region with the disturbed flow. In the following section, we discuss in greater detail the effect of shear stress on endothelial phenotypes.

## 4. The Effect of Shear Stress on the Vascular Endothelium

Endothelial cells play a vital role in cardiovascular homeostasis. Alterations in blood flow characteristics control vascular physiology and pathobiology in both healthy and diseased conditions through hemodynamic forces. The flow-related variations in hemodynamic forces influence endothelial phenotype and control various vascular processes, including the maintenance of acute vessel tone, vascular permeability, leukocyte adhesion, blood vessel development, and secretion of prothrombotic and anti-thrombotic signaling molecules. In straight parts of arteries, the blood flow is laminar and the wall shear stress is high, creating an athero-protective effect and promoting a healthy endothelium [67]. Contrarily, the conditions in branches and curvatures, where blood flow is disturbed with nonuniform and irregular distribution of low wall shear stress, promote atherogenesis and contribute to atherosclerosis, peripheral artery disease, cerebral cavernous malformations (CCMs) [68], and stent re-stenosis and thrombosis [69].

### 4.1. Effects of Shear Stress on Leukocyte Adhesion

The vascular endothelial cell lining serves as a barrier between the blood and tissue to regulate the passage of fluids and migration of leukocytes from the vascular lumen into the vascular tissue [70]. Rolling leukocytes in the bloodstream are recruited to the inflammatory site during oscillatory shear stress and adhere to the vessel wall as a defense mechanism through molecular interaction with immunoglobulin-like adhesion molecules such as VCAM-1, ICAM-1, and E-Selectin [71]. The sequestration of leukocytes in the artery wall is one of the hallmarks of early atherosclerosis. Laminar shear stress limits proinflammatory gene expression and promotes endothelial quiescence with several pathways involving eNOS activation or KLF2 expression. The anti-inflammatory properties of Kruppel-like factor-2 (KLF2) prevent leukocyte adhesion by reducing the expression of VCAM-1 and ICAM-1 [72]. In contrast, oscillatory shear promotes leukocyte adhesion via multiple proinflammatory pathways, eventually leading to NF-κB or Yap activation [73,74]. Apart from the family of CAMs, the intercellular membrane proteins PECAM-1 and JAM-A additionally serve as cell surface receptors and promote leukocyte adhesion and transmigration via LFA-1 binding [75,76]. This flow-dependent increase in the number of cell adhesion receptors in the artery wall promotes leukocyte adhesion and the onset of inflammatory processes [77], which favors atherosclerosis plaque progression and EC dysfunction [78,79].

### 4.2. Effects of Shear Stress on Vascular Permeability

In order to maintain tissue homeostasis, the permeability and structural integrity of the endothelium is controlled by a number of membrane-associated proteins, including JAMs, Claudins, PECAM-1, and VE-cadherin [20]. Endothelial junctions and their molecular organization govern permeability based on the requirements of the organism in both healthy and diseased states [70]. Depending on the pattern of flow, endothelial cells may line the cytoskeleton and induce traction forces to maintain the junctional integrity of cell–cell connections with the help of several proteins in cell–cell junctions, including JAMs, PECAM-1, and VE-cadherin [4]. Laminar flow influences Rho-A activity, cytoskeleton tension, and alignment of cells via actin polymerization and myosin phosphorylation. In contrast, disturbed flow induces hyperactivation of Rho kinase activity, leading to cytoskeletal rearrangement and endothelial barrier dysfunction via the inactivation of myosin phosphatase [80]. Furthermore, disturbed flow induces PECAM-1–VE-cadherin–VEGFR2 complex signaling to drive ECM remodeling and increase fibronectin expression, which play critical roles in inflammatory activation. This membrane-protein-dependent mechanotransduction pathway activates p21-activated kinase (PAK), NF-κB, JNK, and others, degrading junctional integrity and increasing paracellular permeability, which induces greater leakage across the barrier. This matrix-dependent activation of PAK suppresses myosin light-chain kinase and causes a reduction in the phospho myosin light-chain [81]. Additionally, these PAK-dependent modifications of VE-cadherin and occludin phosphorylation lead to their internalization [82].

### 4.3. Effects of Disturbed Flow on EndMT

EndMT is essential for embryonic development, normal capillary vessel formation, smooth muscle cell differentiation, and angiogenesis [83]. However, it is also a critical player in many pathologies, including atherosclerosis, fibrosis, cancer progression, and chronic inflammation [83,84]. The transition of endothelial cells into mesenchymal cells contributes substantially to the progression of atherosclerosis through the control of inflammation, the structural integrity of the plaque, and the synthesis of the extracellular matrix and collagen. Recent studies also showed that in ECs exposed to oscillatory shear stress, FN and integrin β1 signaling induced upregulation of the TGF-β receptor and activated several signaling pathways, primarily the Smad 2/3 pathway. Furthermore, disturbed flow-induced phospho-Smad 2/3, and its nuclear localization increased inflammatory gene markers such as SNAI1, NOTCH3, and FN and promoted vascular remodeling. Blocking this pathway has reduced atherosclerosis in in vivo studies [31,85]. Inflammatory mediators also aid in the development of EndMT by stimulating NF-κB [86]. Laminar flow promotes the endothelial phenotype through KLF4 target gene induction. Recently, Tenascin-x (TN-X) has been shown to be a critical modulator of flow-induced inhibition of EndMT, endothelial inflammation, and atherogenesis under laminar flow in a KLF2-dependent manner [87].

### 4.4. Effects of Disturbed Flow on Glucose Metabolism

Recently, cellular metabolism has been shown to regulate the EC phenotype. Under inflammatory conditions, EC activation and dysfunction-driven metabolic reprogramming are key contributors to the pathophysiology of many diseases, such as atherosclerosis, diabetic retinopathy, pulmonary hypertension, and cancer. As described above, laminar shear stress promotes quiescent ECs through the expression of a range of transcriptional factors, including (KLF2), which promotes anti-inflammatory and anti-thrombotic properties. Similarly, laminar flow induction of KLF2 is known to regulate cellular metabolism by repressing the expression of the glycolytic regulator phosphofructokinase-2/fructose-2,6-bisphosphatase-3 (PFKFB3) to maintain quiescent ECs. In disturbed flow conditions, the loss of KLF2 function can suppress PFKFB3, which enhances glycolysis, inflammatory responses, and insulin resistance [88]. In addition, oscillatory shear stress promotes ROS-induced stabilization of hypoxia-inducible factor (hif-1 alpha), which acts as a major transcription factor stimulating the NF-κB pathway and a subsequent increase in glucose regulators such as PFKFB3, HK2, and GLUT1 [89]. Recent studies demonstrated that inflammatory mediators reprogram cellular metabolism and enhance glycolysis, mitochondrial oxidative phosphorylation (OXPHOS), and the pentose phosphate pathway via NF-κB-PFKFB3 signaling, which promotes further inflammation. Blocking this pathway by silencing PFKFB3 reduced inflammation in an in vivo model [90]. During hypoxia and angiogenesis, to maintain the cellular energy requirements, ECs may regulate glucose utilization via PFKFB3 activation. However, hyperactivation of PFKFB3 and NF-κB signaling results in inflammatory progression due to increased monocyte adhesion, permeability, and EndMT transition, which can lead to various vascular inflammatory diseases.

Figure 1 shows a schematic diagram representing the changes occurring in an endothelial cell upon exposure to oscillatory shear stress. An increase in fibronectin expression due to disturbed flow mediates integrin signaling and activates the NF-κB, PAK, and JNK pathways and YAP activation, which increases inflammation via the upregulation of cell adhesion molecules and leukocyte adhesion. PECAM-1, VE-Cadherin, and the VEGF complex drive this extracellular matrix remodeling and increase fibronectin expression. Activation of PAK and an increase in Rho kinase activity due to disturbed flow induces VE-cadherin, occludin internalization, cytoskeletal remodeling, and myosin light-chain contraction, which creates a gap in the junction and promotes leukocyte transmigration. The flow-induced activation of multiple signaling pathways promotes Smad 1/5 phosphorylation and induces EndMT. KLF2 promotes suppression of all these effects in laminar flow. KLF2 reduction and PFKFB3 upregulation induce all these adverse effects in disturbed flow.

## 5. Vascular Inflammatory Diseases

### 5.1. Atherosclerosis

Atherosclerosis is a chronic inflammatory disease. A healthy endothelium actively reduces thrombosis, vascular inflammation, and hypertrophy in addition to mediating endothelium-dependent vasodilation. Endothelial dysfunction and inflammation are responsible for various pathologies involved in atherosclerosis.

#### 5.1.1. Plaque Instability and Endothelium

Recent clinical data indicate that plaque disruption rather than plaque stenosis is related to cardiovascular events and mortality in patients [91]. Unstable plaques feature a large necrotic core, high macrophage content, and thin fibrous cap [92]. Evidence indicates plaque destabilization is caused by inflammation due to immune cell infiltration in atherosclerotic lesions. Matrix degradation and apoptosis, two crucial factors in plaque stability, are controlled by inflammatory mediators from macrophages and T cells. Owen’s group recently revealed that ACTA2+ myofibroblast-like cells originate not only from smooth muscle cells but also from endothelial cells and macrophages via EndMT and macrophage-to-mesenchymal transition (MMT), respectively [93]. Another study supports that control of EndMT could be a valid strategy for plaque stabilization. Kovacic et al. examined histone post-translational modification during EndMT and found that histone deacetylation by HDAC9 is increased. Inhibition of class IIa HDAC family members, blocked-EndMT-mediated gene induction, and endothelial HDAC9 knockout reduced atherosclerosis and enhanced plaque stability [94]. OCT4, a Yamanaka factor, was recently found to be activated in ECs during atherogenesis and plays an important role in plaque stability. Endothelial deletion of OCT4 exacerbated atherosclerosis in ApoE-null mice plaque stability by regulating endothelial ABCG2 induction to control excessive hemes and ROS [95]. It has been observed that oscillatory shear stress enhances the activity of metalloproteinase (MMP), drives collagen degradation, and weakens fibrous caps, in addition to its role in vascular remodeling with matrix degradation [96,97]. A recent in vivo investigation on coronary arterial plaque clearly demonstrated that the low shear stress area was highly correlated with macrophages, cholesterol crystals, 18F-NaF activity, active microcalcification, and thin-cap fiberoatheroma (TCFA) thinning [98]. Proinflammatory signaling and anerobic metabolism are tightly correlated with atherosclerotic macrophages through the involvement of HIF-1 and PFKFB3. High PFKFB3 expression in mice is linked to an unstable plaque, whereas suppression of PFKFB3 activity stabilizes the plaque [99,100,101,102]. Many recent studies have proven that fluid dynamics and endothelial shear stress are the main pathological factors leading to vulnerable plaques [103].

#### 5.1.2. Plaque Calcification and Endothelium

Intimal calcification is associated with atherosclerosis. Unstable plaques are at risk of rupture due to the involvement of inflammation and microcalcification, which are controlled by wall shear stress [104,105,106,107]. Early proinflammatory and osteogenic cytokines from macrophages including TNF-𝛼, IGF-1, TGF-β, IL-1β, IL-6, and IL-8 induce differentiation of the vascular smooth muscle cells into osteoblast-like cells, initiating microcalcification [108]. TNF-𝛼 induces a reduction in BMPR2, enhancing BMP-9 for osteogenic differentiation in ECs, and induces calcification, thin fibrous cap formation, and plaque rupture [96,97,109,110,111]. Vascular calcification is inhibited by KLF2-mediated suppression of endothelial BMP/SMAD1/5 signaling resulting from laminar flow [112]. The metabolic change from OXPHOS to aerobic glycolysis and downregulation of PPAR-𝜰 resulting from enhanced WNT/β-catenin pathway activation in atherosclerotic lesions increases vascular calcification via lactate secretion. Inhibition of glycolysis by 3-PO, the PFKFB3 inhibitor, reduces EC differentiation and VSMC calcification and enhances cell survival [113,114,115,116]. Recent studies demonstrated that FGF21 reduces vascular calcification and improves vascular function by increasing antioxidant SOD and reducing oxidative stress [117,118].

### 5.2. Pulmonary Arterial Hypertension (PAH)

The hallmarks of the devastating condition known as pulmonary arterial hypertension (PAH) are neointimal lesions, small vessel constriction, and large vessel stiffness. Pulmonary arterial hypertension occurs when most of the small arteries throughout the lungs narrow in diameter, which leads to increased pulmonary vascular resistance, subsequent right heart failure, and premature death [119]. Endothelial dysfunction is a key factor in the onset and development of artery remodeling and disease progression in pulmonary arterial hypertension (PAH) [120,121,122]. EC dysfunction can be caused by several factors, including inflammatory cytokines, hypoxia, toxins, and external stimuli such as shear stress. Among them, oscillatory shear stress, which leads to disturbed flow, fluid dynamics, and ongoing pathogenic mechanisms, has a substantial impact on PAH patients with neointimal lesion formation, leading to vessel remodeling, which increases pressure in the blood flow [119,123]. PAH begins with a malfunctioning EC, progresses with increased cell proliferation and decreased apoptosis, and finally matures with senescence, which makes PAH difficult to reverse. BMPR2 mutation is frequently found in familial PAH and idiopathic PAH patients. Endothelial BMPR2 controls TGF-β-dependent EndMT, and BMPR2 ligand BMP9 administration has been used to reverse PAH in a rat model. Pulmonary inflammation is believed to put patients with BMPR2 mutations at risk of PAH development [122]. Single-cell RNA sequencing of endothelial cells from a PAH mouse model showed that ECs upregulate genes in the MHC class II pathway, supporting the role of ECs in the inflammatory responses in PAH [124]. A recent report by Chan et al. indicated endothelial senescence is another factor driving PAH [125]. Induction of endothelial senescence was induced by blocking fraxacin (FXN) and mitochondria iron–sulfur (Fe–S) cluster assembly protein, and PAH patients showed reduced fraxacin expression and EC senescence. In addition, senolytic treatment prevented FXN-dependent PAH development in mice. Shyy et al. reported that the transcriptional coactivator MED1 is downregulated in PAH patients and animal models. MDE1 is associated with KLF4, an important transcriptional regulator in endothelial homeostasis, and mediates the expression of BMPR2, ETF, and TGFBR2 [126]. Several studies have demonstrated that disturbed blood flow initiates PAH with EC inflammation, barrier dysfunction, and smooth muscle cell migration through downregulation of eNOS / AKT pathway [127]. Metabolic alterations such as aerobic glycolysis are involved in the development of PAH. Inhibition of the glycolytic enzyme PFKFB3 inhibited PAH progression by reducing vascular remodeling, endothelial inflammation, and leukocyte recruitment [128,129,130]. BMPR2 mediated NOTCH1 activation [131] and increased the nuclear localization of HDAC4 and HDAC5, which are involved in the promotion of EC proliferation and a reduction in apoptosis in PAH ECs [132].

### 5.3. Sepsis

Sepsis is caused by uncontrolled immune responses to bacterial or viral infections, damaging host tissues. Endothelial proinflammatory phenotype changes are responsible for many sepsis-induced pathological events, including vascular permeability, increased coagulation, and hypoperfusion [133].

#### 5.3.1. Vascular Leakage

Sepsis-induced vascular leakage leads to fluid accumulation and multiorgan failure. A recent study on the ProCESS Trial showed that endothelial permeability markers (sFLT-1, Ang-2, and VEGF) were significantly lower in number in sepsis survivors [134]. Endotoxins or viral pathogens cause vascular leakage via EC junction disassembly or cell death. VE-cadherin is a key adherens junction protein, and its disengagement leads to paracellular permeability. Tyrosine phosphorylation of VE-cadherin induces its intracellular relocalization and loss of cell–cell adhesion. Inflammatory ligands activate Rho-dependent actin stress fiber formation and contractility and paracellular pore formation. Evidence shows that the blockade of endothelial permeability can be a valid strategy for decreasing sepsis mortality [135,136]. Recently, Okada et al. revealed a signaling pathway in ROBO4-dependent endothelial barrier stabilization using small-molecule screening [137]. Endothelial TGF-β-ALK5-Smad2/3 signaling led to ROBO4 expression, whereas BMP9-ALK1-Smad1/5 signaling blocked it, and the ALK1 inhibitor suppressed vascular hyperpermeability and mortality in COVID-19 mouse models. Lactate is known to be a sepsis biomarker, and its newly identified effect on the endothelial barrier has been reported. According to a report by Li et al., lactate disorganizes the VE-cadherin complex via VE-cadherin cleavage and internalization via calpain and Erk2 activation [138].

#### 5.3.2. Thrombosis

Sepsis is often associated with subclinical hypercoagulability or acute disseminated intravascular coagulation (DIC) accompanied by widespread microvascular thrombosis and consumption of platelets and coagulation proteins, leading to bleeding [139]. Sepsis induces a shift in the EC phenotype toward a procoagulant and anti-fibrinolytic status. Activated ECs, in the presence of proinflammatory molecules, express tissue factor (TF), which binds with circulating coagulation factor VII in the extrinsic pathway of the blood coagulation system. Activated ECs promote platelet adhesion and aggregation via the upregulation of various cell surface adhesion molecules [140]. During sepsis, PAI-1(plasminogen activator inhibitor-1) is released by ECs, leading to the inhibition of fibrinolysis and hypercoagulation.

#### 5.3.3. Vascular Tone

Sepsis leads to high cardiac output and decreased peripheral resistance due to vessel dilation and hypotension. Inflammatory ligands cause increased NO synthesis from ECs and smooth muscle cell relaxation [141]. NO also reacts with ROS to form nitrate and nitrate-inducing cytotoxicities [142].

Proinflammatory stimuli such as diabetes, hypercholesterolemia, disturbed flow, ECM remodeling, and inflammatory cytokines induce activation of ECs and modification of phenotypes via the upregulation of inflammatory proteins and activation of several signaling pathways. This proinflammatory phenotype causes EC dysfunction, leading to weakening of the barrier, an increase in permeability, an increase in inflammation, EndoMT, senescence, etc, which is shown in Figure 2. These factors are the main causes of the development of most cardiovascular diseases.

Cao, Y. et al. [128]Chen, M. et al. [143]Culley, M.K. et al. [125]Liang, G. et al. [87]Morita, M. et al. [137]Wang, C. et al. [130]Wang, C. et al. [126]Yang, K. et al. [138]Yun, S. et al. [63]

## 6. Conclusions

### 6.1. Targeting ECs in Inflammatory Diseases

Endothelial activation or dysfunction is accompanied by a number of chronic inflammatory diseases and plays a pivotal role in disease initiation and progression. However, in the past, endothelial cells have not been the main therapeutic targets in inflammatory diseases. For example, the cytokine storm induced by COVID-19 infection caused hyperactivation of inflammatory cytokines, which induced the destruction of endothelial cell–cell junction and vascular leakage, leading to fatal organ dysfunction. A number of clinical trials targeting inflammatory cytokines have not found it to be a successful method for sepsis treatment, presumably due to the heterogeneous nature of pathogens, the need for inflammation in pathogen clearance, and so on. Recent results suggest that promoting endothelial barrier function can markedly decrease sepsis mortality [135,136,144]. Targeting endothelial cells in inflammatory diseases has the additional advantage of reducing the risk of infection due to the compromised immune function from systemic anti-inflammatory treatments. We identified a clinically available compound that can strengthen endothelial adherens junction without affecting leukocyte activation. The compound successfully alleviated acute lung injury in a sepsis mouse model (unpublished data). Psoriasis is a chronic inflammatory skin disease with erythematous plaques. Psoriatic skin manifests high angiogenesis in lesions, and recent reports suggest that blockade of angiogenesis suppresses psoriatic plaque development. Li et al. identified an EC subtype with higher IGFBP7 expression in psoriasis patient skin lesions, which promoted psoriasis via destruction of the endothelial glycocalyx, and antibody-mediated blockade of IGFBP7, which attenuated T-cell adhesion on EC layers in a mouse psoriasis model [145]. Identifying a disease-specific endothelial feature (high levels of EC-secreted IGFBP7 in psoriasis) and its utilization for therapeutic intervention is a very promising strategy since it does not affect the normal EC population.

### 6.2. Identifying Disease-Specific EC Phenotypes or EC Subpopulations

Endothelial cells often induce phenotypic changes in response to various stresses (such as infection, disturbed blood flow, dyslipidemia, and dysglycemia), and when the adaptation process is prolonged and other disease-inducing factors are combined, the EC phenotypes promote the development of vascular diseases. Reversing EC phenotype changes proved to be a useful therapeutic strategy in the STELLAR trial for a PAH drug, sotatercept, targeting endothelial TGF-β signaling and EndMT [146]. A recently developed scRNA sequencing technique enabled researchers to delineate endothelial phenotypes in extreme detail. Carmeliet’s group identified a new EC subtype that is downregulated in tumors compared to neighboring normal tissues [147]. This new EC subpopulation, called LIPECs (lipid-processing ECs), has higher expression of PPAR-γ pathway genes. Retrospective analysis of a patient dataset revealed that breast cancer patients treated with metformin had reduced mortality and an increased LIPEC population. Disturbed flow has a major impact on endothelial phenotypes, and Jo et al. recently performed a single-cell RNA sequencing analysis by inducing disturbed flow in mouse aortae using PCL (partial carotid ligation) [148]. They found that, under proatherogenic disturbed flow, endothelial cells not only undergo EndMT but also undergo a new phenotypic change to immune-cell-like EC subtypes via a process called EndICLT (endothelial–immune-cell-like transition). Identifying the role of this EC subtype in flow signaling could lead to a novel therapeutic target for atherosclerosis treatment.

In conclusion, endothelial cells deserve more attention considering their active role in many vascular inflammatory diseases, and an enhanced understanding of EC phenotypes using recently developed advanced NGS technology would allow us to specifically target endothelial-disease-promoting features with fewer side effects.

## Figures and Tables

**Figure 1 cells-12-01640-f001:**
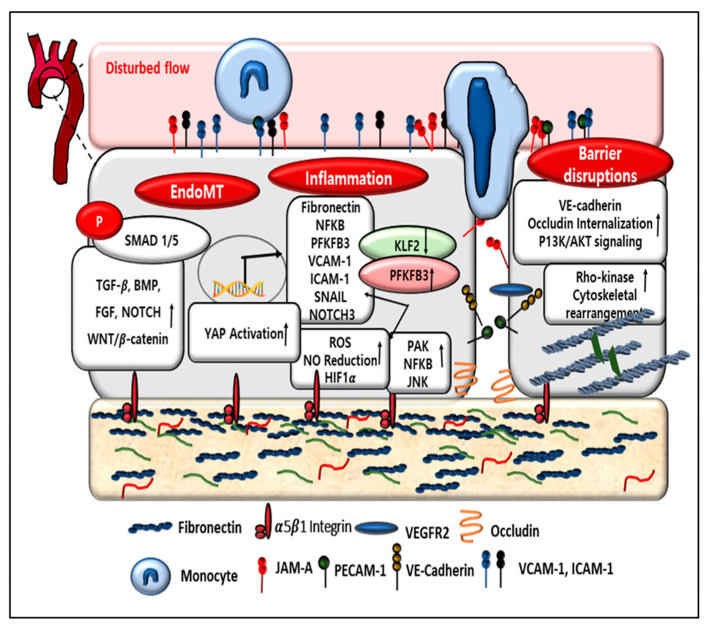
Effects of oscillatory shear stress on vascular endothelium.

**Figure 2 cells-12-01640-f002:**
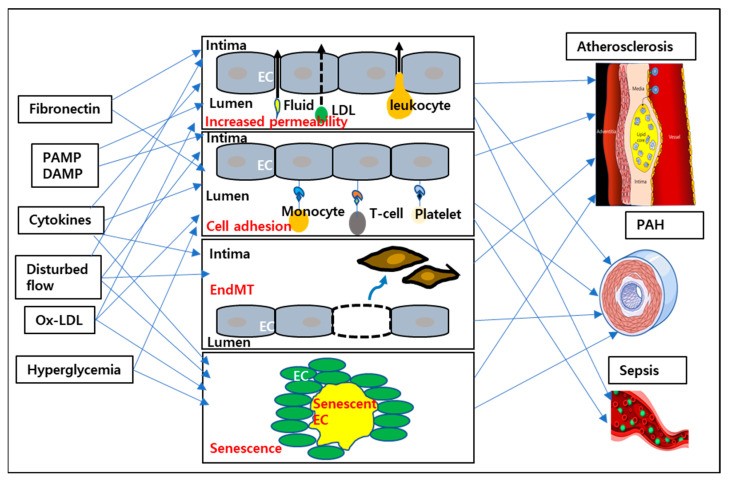
EC dysfunction and the progression of vascular diseases.

## Data Availability

Not applicable.

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
