# Peer review of "Vascular Inflammatory Diseases and Endothelial Phenotypes"

_cells, 2023, doi:10.3390/cells12121640_

Round 1
Reviewer 1 Report
The researchers aimed to review the endothelial cells' physiology modification associated with the signaling pathways induced by the inflammatory process in vascular diseases. The article emphasizes the endothelium pro-inflammatory phenotype by presenting adhesion molecules signaling pathways, the process of transition from endothelial cells to mesenchymal cells, and senescence. In my opinion, in this section, the presentation is much too general, and the opinion of researchers on the changes occurring in the phenotype of endothelial cells is not presented.
The pro-inflammatory and stress stimuli and their role in modifying the phenotype of endothelial cells, in correlation with the different pathologies that these changes can cause, are very well presented.
The main significant vascular inflammatory pathologies are well presented in correlation with the inflammatory process's effect on endothelial cells' phenotype.
The authors should address my following comments and suggestions to bring the manuscript to the standard of the Cell journal:
1. The clarity of images could be higher. Please provide images with higher clarity and magnification
2. The grammar is often not the most appropriate. The text contains phrases without a verb or words whose clarity is not very good.
3. In the text, the authors use abbreviations without explanation. In addition, as far as I know, we cannot use abbreviations in the Abstract. Some of the abbreviations are not correct. E.g., TNFa, instead of TNFα
Author Response
Response to Reviewer 1 Comments
- The clarity of images could be higher. Please provide images with higher clarity and magnification
We’ve improved the clarity of images and magnified the size of font.
- The grammar is often not the most appropriate. The text contains phrases without a verb or words whose clarity is not very good.
We’ve revised the text and removed the grammatical and syntax errors and language-edited the text using a professional service.
- In the text, the authors use abbreviations without explanation. In addition, as far as I know, we cannot use abbreviations in the Abstract. Some of the abbreviations are not correct. E.g., TNFa, instead of TNFα
Thanks, the abbreviations have been removed in the abstract. The abbreviations used were corrected as suggested.
Reviewer 2 Report
This is a very broad-spectrum review and is an excellent paper for readers not expert in the field but looking for a comprehensive introduction to it. It is well organised, update and with a good set of references.
It will make a very good introductions to the special issue on "Vascular Endothelial Functions in Health and Diseases" and I will certainly refer to it in future.
A check of the language for occasional minor problems may be useful.
Author Response
This is a very broad-spectrum review and is an excellent paper for readers not expert in the field but looking for a comprehensive introduction to it. It is well organised, update and with a good set of references.
It will make a very good introductions to the special issue on "Vascular Endothelial Functions in Health and Diseases" and I will certainly refer to it in future.
A check of the language for occasional minor problems may be useful.
Thanks a lot! We’ve revised the text and removed the grammatical and syntax errors and also had our manuscript English-edited by a professional service. Thanks.
Reviewer 3 Report
This review discusses pro-inflammatory stimuli and their effects on endothelial cell function in the context of cardiac diseases.
As the title says: Vascular inflammatory diseases and endothelial phenotypes but MS does not adequately cover these two parameters. EdMT is poorly explained, and there is no molecular mechanism elaboration. The authors presented the EC with a good amount of information, however the review lacks direction. They could better explain the mechanism of EC in vascular diseases. They are discussing inflammation in more detail, but the disease specific approach is unclear. A simple orientation of this review will make it easier to read and informative. I would suggest choosing disease condition first and then discussing inflammation, and the specific role of EC. For example, plaque formation? P3L101: please elaborate more on plaque formation if it is associated with EC. Why plaque formation at coronary arteries is more common than at other arteries?
Refer to P1L24: Please confirm if it is a Heterogeneous or heterogenous?
Overall, the review is too cumbersome to read, and the author simply dumps everything in one place.
Authors should revise the text to remove any grammatical and syntax errors. By doing so, readers will have a better understanding of the information presented.
Author Response
Response to reviewer comment:
Thank you for the reproductive comments. We’ve added molecular mechanims of EndMT in line 103 through line 120. We’ve added some orientation paragraphs or sentences with a direction of the review in line 50-53, line 59-90, line 117-118, line 197-198. We’ve reorganized Section 5. Vscular inflammatory diseases according to your suggestion by removing ‘diabetes and atheroclerosis’ and ‘EndMT and atherosclerosis’. The section looks more focussed on diseases and EC function. We’ve added another sentence to extend the describtion P3L101 in line 92. We’ve revised the text and removed the grammatical and syntax errors and language-edited the text using a professional service.
Reviewer 4 Report
This review entitled “Vascular inflammatory diseases and endothelial phenotypes” focused on the recent research on endothelial cells in inflammation, specifically addressing their relevance to cardiovascular disease. While the topic is interesting and holds potential value for advancing treatments of cardiovascular disease, there are concerns regarding the structure as well as the insights presented in this review. I would advise addressing the following concerns before considering this article for publication in Cells.
Specific comment 1: Structure
The authors covered a broad aspect of endothelial properties and inflammatory stimuli while spending limited sentences explaining each point. This makes the first 4 pages appear as a bullet point summary of endothelial characteristics rather than a cohesive review. I would advise the authors to utilize the first 3 sectors (introduction, pro-inflammatory phenotypes, pro-inflammatory stimuli) to lay the background and framework for the later discussion on shear stress and inflammatory vascular disease. For example, instead of listing individual cytokines, the author might want to group them based on types of endothelial activation (See Jordan S. Pober and William C. Sessa, nature review, 2007).
In addition, the authors should also use this chance to ensure the comprehensiveness of the introduction. For example, in line 29, the authors introduced endothelial permeability as regulated by intercellular junctions. However, endothelial permeability to substrates and immune cells is also affected by transcytosis (Chenghua Gu lab, Harvard) and transcellular migration (Sarah E. Lutz lab, UIUC). To strengthen this review article, the authors may expand the reference list.
Specific comment 2: Insights and clinical relevance
The author presented knowledge about share stress, endMT, and vascular diseases such as atherosclerosis. Despite a clear description of these potential mechanisms, the author’s insights on how this could inform further research of the field and clinical development are not clear to me.
Specific comment 3: Introduction
Context and phrases, especially in the first paragraph, are common across manuscripts with similar topics. It could be a great opportunity for the authors to highlight the uniqueness and significance of this paper.
See above
Author Response
Response to Reviewer 4 Comments
This review entitled “Vascular inflammatory diseases and endothelial phenotypes” focused on the recent research on endothelial cells in inflammation, specifically addressing their relevance to cardiovascular disease. While the topic is interesting and holds potential value for advancing treatments of cardiovascular disease, there are concerns regarding the structure as well as the insights presented in this review. I would advise addressing the following concerns before considering this article for publication in Cells.
Specific comment 1: Structure
The authors covered a broad aspect of endothelial properties and inflammatory stimuli while spending limited sentences explaining each point. This makes the first 4 pages appear as a bullet point summary of endothelial characteristics rather than a cohesive review. I would advise the authors to utilize the first 3 sectors (introduction, pro-inflammatory phenotypes, pro-inflammatory stimuli) to lay the background and framework for the later discussion on shear stress and inflammatory vascular disease. For example, instead of listing individual cytokines, the author might want to group them based on types of endothelial activation (See Jordan S. Pober and William C. Sessa, nature review, 2007).
In addition, the authors should also use this chance to ensure the comprehensiveness of the introduction. For example, in line 29, the authors introduced endothelial permeability as regulated by intercellular junctions. However, endothelial permeability to substrates and immune cells is also affected by transcytosis (Chenghua Gu lab, Harvard) and transcellular migration (Sarah E. Lutz lab, UIUC). To strengthen this review article, the authors may expand the reference list.
Thanks for the very helpful comments. According to your suggestion, we’ve changed the section ‘3.1 inflammatory cytokines’ section avoiding bullet point summary format and grouped cytokines depending on types of endothelial activation. For cohesiveness, we’ve added paragraphs or sentences for the direction of the review as in line 52-60, line 120-130, line 228-230.
Transcellular migration and transcytosis and their role in CNS endothelial cells have been described in line 84-94 with more references.
Specific comment 2: Insights and clinical relevance
The author presented knowledge about share stress, endMT, and vascular diseases such as atherosclerosis. Despite a clear description of these potential mechanisms, the author’s insights on how this could inform further research of the field and clinical development are not clear to me.
We thank you and totally agree with your points. We’ve re-written the Conclusions section completely to provide insights and clinical relevance of targeting endothelial cells for diseases.
Specific comment 3: Introduction
Context and phrases, especially in the first paragraph, are common across manuscripts with similar topics. It could be a great opportunity for the authors to highlight the uniqueness and significance of this paper.
Thanks. We’ve highlighted the significance of the paper in line 50-55.
Reviewer 5 Report
This review is an extensive compilation of articles on cardiovascular disease and vascular endothelial cell lesions. Particular attention is given to the molecules responsible for vascular endothelial cell lesions. It is expected to contribute to the development of diagnostic and therapeutic methods for a variety of diseases.
I have no concerns about its content. I would like to point out the following issues. Appropriate corrections are requested.
Incomplete or incorrect sentences are scattered mainly in the first half of the manuscript. e.g. lines 61, 63,72,83,117,133,203,262,317,404
There are some inconsistencies in abbreviations of molecules and genes. such as. IL1b IL-1b, IL6 IL-6
Some abbreviation of responses such as EndMT, TCFA first appeared without full spelling
The letters in Figures 1 and 2 are too small to understand.
As above
Author Response
This review is an extensive compilation of articles on cardiovascular disease and vascular endothelial cell lesions. Particular attention is given to the molecules responsible for vascular endothelial cell lesions. It is expected to contribute to the development of diagnostic and therapeutic methods for a variety of diseases.
I have no concerns about its content. I would like to point out the following issues. Appropriate corrections are requested.
Incomplete or incorrect sentences are scattered mainly in the first half of the manuscript. e.g. lines 61, 63,72,83,117,133,203,262,317,404
There are some inconsistencies in abbreviations of molecules and genes. such as. IL1b IL-1b, IL6 IL-6
Some abbreviation of responses such as EndMT, TCFA first appeared without full spelling
Response to reviewer comments
The letters in Figures 1 and 2 are too small to understand.
Thank you for your kind comments. We’ve revised the text and removed the grammatical and syntax errors. Abbreviations have been checked and unified throughout the manuscript. E.g.,( please refer lines from 143-179 ). Explanation for the abbreviations has added. E.g., (line 69-71) We’ve Improved the clarity of images and magnified the size of font.
Reviewer 6 Report
The review manuscript by Immanuel and Yun with a title “Vascular inflammatory diseases and endothelial phenotypes” provides an interesting overview of the current literature on endothelial cells and the vasculature in inflammatory diseases. The review is well structured. After a brief introduction, the endothelial proinflammatory phenotypes, EC-proinflammatory stimuli, shear stress on the vascular endothelium and vascular inflammatory diseases are described. The overall structure of the review is concise and clear. The two figures, which illustrate proinflammatory processes and diseases of the vascular system , are well integrated in the text and clearly illustrate the text content. The authors make it clear that endothelial plasticity and endothelial cell-specific inflammatory mechanisms are important mechanisms of various systemic diseases and should be considered in therapeutic approaches. The review has some shortcomings, which are listed below. These are mostly minor errors that can be improved before the publication.
Major: intensive language editing should be carried out before publication
Figure 1 and Figure 2: the font size should be increased in this figure, as it is not legible in the present form. All abbreviations should be explained so that the figure can be understood without reading the text.
Figure 2: I would delete the citations from this figure.
Minor:
Line 62: “EC phenotypic changes during vascular inflammation include;”- I would finish that sentence e.g. “EC phenotypic changes during vascular inflammation are discussed below”.
Line 71: “JAM-A, JAM-B, JAM-C, Ig superfamily adhesion receptors” This sentence is missing a verb.
Line 74: This sentence is missing a verb.
Line 78 “Chemokine:” I would use “chemokines” instead while we always speak about multiple chemokines
Line 81: “MCP-1(CCL2) binds CCR2 on monocytes. Ox-LDL or disturbed flow induces MCP-1 expression on vascular wall. Functions in recruitment of monocytes to atheroma16.” This sentence doesn’t make any sense. Please rephrase.
Line 116: “IL1b: produced by monocyte, macrophage and EC29.” I would use the plural: “monocytes, macrophages and ECs”
please unify this throughout the manuscript: “NFKB” and “NFkB”-
“TGF-b“and ”TGF-B” or “TGF-b” or “TGF-beta”
“TNF-a”, “TNF-a”
“COVID19”, “COVID-19”
Please introduce the abbreviation on first use, e.g. TGF-b is introduced in line 255 but has been used previously without explanation (e.g. line 105)
Figure 1 caption: “occluding” should be “occludin”.
The abbreviation “MMP” is introduced twice and differently
“FN” and “FN1” the abbreviation for fibronectin should be unified and introduced only once
the quality of English Language is addressed in the comments for authors.
Author Response
Major: intensive language editing should be carried out before publication
Figure 1 and Figure 2: the font size should be increased in this figure, as it is not legible in the present form. All abbreviations should be explained so that the figure can be understood without reading the text.
Figure 2: I would delete the citations from this figure.
Minor:
Line 62: “EC phenotypic changes during vascular inflammation include;”- I would finish that sentence e.g. “EC phenotypic changes during vascular inflammation are discussed below”.
Line 71: “JAM-A, JAM-B, JAM-C, Ig superfamily adhesion receptors” This sentence is missing a verb.
Line 74: This sentence is missing a verb.
Line 78 “Chemokine:” I would use “chemokines” instead while we always speak about multiple chemokines
Line 81: “MCP-1(CCL2) binds CCR2 on monocytes. Ox-LDL or disturbed flow induces MCP-1 expression on vascular wall. Functions in recruitment of monocytes to atheroma16.” This sentence doesn’t make any sense. Please rephrase.
Line 116: “IL1b: produced by monocyte, macrophage and EC29.” I would use the plural: “monocytes, macrophages and ECs”
please unify this throughout the manuscript: “NFKB” and “NFkB”-
“TGF-b“and ”TGF-B” or “TGF-b” or “TGF-beta”
“TNF-a”, “TNF-a”
“COVID19”, “COVID-19”
Please introduce the abbreviation on first use, e.g. TGF-b is introduced in line 255 but has been used previously without explanation (e.g. line 105)
Figure 1 caption: “occluding” should be “occludin”.
The abbreviation “MMP” is introduced twice and differently
“FN” and “FN1” the abbreviation for fibronectin should be unified and introduced only once
We’ve improved the clarity of images and magnified the size of font. The abbreviations has been explained. Citations from the figure 2 has been removed. Revised the text and removed the grammatical and syntax errors. E.g.,( please refer lines from 71-169 ) Abbreviations have been checked and unified throughout the manuscript. E.g.,( please refer lines from 144-170 ) Explanation for the abbreviations has done on first use. All the minor mistakes mentioned has been corrected.
Thanks.
Round 2
Reviewer 3 Report
No more comments.
Reviewer 4 Report
The revisions have significantly strengthened this article, and I believe it is now suitable for acceptance.